# Universal Length Generalization with Turing Programs

Kaiying Hou [1 2]   David Brandfonbrener [3]   Sham Kakade [3]   Samy Jelassi [* 4]   Eran Malach [* 3]

## Abstract

Length generalization refers to the ability to extrapolate from short training sequences to long test sequences and is a challenge for current large language models. While prior work has proposed some architecture or data format changes to achieve length generalization, these proposals typically apply to a limited set of tasks. Building on prior scratchpad and Chain-of-Thought (CoT) techniques, we propose *Turing Programs*, a novel CoT strategy that decomposes an algorithmic task into steps mimicking the computation of a Turing Machine. This framework is both universal, as it can accommodate any algorithmic task, and simple, requiring only copying text from the context with small modifications. We show that by using Turing Programs, we obtain robust length generalization on a range of algorithmic tasks: addition, multiplication and in-context SGD. We then demonstrate that transformers achieve length generalization on random Turing Programs, suggesting that length generalization is possible for any algorithmic task. Finally, we theoretically prove that transformers can implement Turing Programs, constructing a simple RASP (Weiss et al. (Weiss et al., 2021)) program that simulates an arbitrary Turing machine.

## 1. Introduction

Transformer-based language models have shown impressive abilities in natural language generation, reading comprehension, code-synthesis, instruction-following, commonsense reasoning, and many other tasks (Brown et al., 2020; Chen

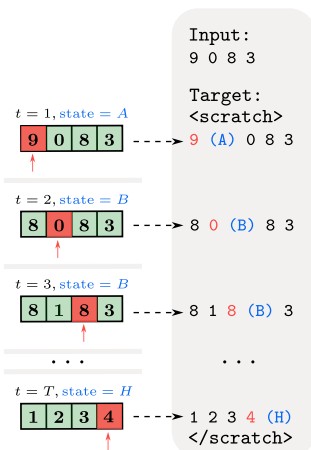

*Figure 1.* Turing Program example for simulating a Turing Machine with scratchpad.

et al., 2021; Chowdhery et al., 2023; Lewkowycz et al., 2022; Gunasekar et al., 2023; Touvron et al., 2023). Despite these impressive abilities, transformers struggle with *length generalization*, which refers to the ability to generalize to longer sequences than seen during training (Abbe et al., 2023; Anil et al., 2022; Jelassi et al., 2023; Zhou et al., 2023). This limitation raises a central question about transformers: are they capable of actually learning an algorithm or do they solve algorithmic tasks by resorting to memorization or shortcuts (Liu et al., 2022)?

Recently, several works have reported poor length generalization of transformers on a wide range of algorithmic tasks (Anil et al., 2022; Delétang et al., 2022; Dziri et al., 2024; Zhang et al., 2022). In parallel, a myriad of papers (Jelassi et al., 2023; Kazemnejad et al., 2024; Shen et al., 2023; Zhou et al., 2023; 2024) have optimized the data formats choice (see section 3 for details) to improve the length generalization of transformers when trained to perform multi-digit addition of two numbers. While the recent progress is impressive—(Zhou et al., 2024) achieve almost perfect accuracy on addition with 100-digit operands while trained on 40-digit, all these "tricks" are specific to the case of addition and may not generalize to other tasks. In contrast, our goal is to develop a technique that is *general* enough to enable length generalization on *any* algorithmic task.

---

*Equal senior contribution. [1]Department of Mathematics, Harvard University, Cambridge, MA, United States [2]Department of Mathematics, UC Berkeley, Berkeley, CA, United States [3]Kempner Institute, Harvard University, Cambridge, MA, United States [4]Center of Mathematical Sciences and Applications, Harvard University, Cambridge, MA, United States. Correspondence to: Kaiying Hou <kaiying@berkeley.edu>.

*Proceedings of the 42^{nd} International Conference on Machine Learning*, Vancouver, Canada. PMLR 267, 2025. Copyright 2025 by the author(s).

*Table 1.* Length generalization results on various problems with Turing Programs. We use $x \to y$ to denote training on $n = x$ and generalizing to $n = y$.

| Problem | Generalization | Accuracy |
|---------|----------------|----------|
| **Addition** $(n + n)$ | $50 \to 100 \, (\mathbf{2}\times)$ | 98% |
| **Multiplication** $(n \times 1)$ | $50 \to 100 \, (\mathbf{2}\times)$ | 97% |
| **Multiplication** $(n \times 3)$ | $50 \to 100 \, (\mathbf{2}\times)$ | 97% |
| **SGD** ($n$ **examples**) | $50 \to 80 \, (\mathbf{1.6}\times)$ | 95% |

To achieve this, we introduce *Turing Programs*, a novel scratchpad technique that may be applied to general algorithmic tasks. This technique is motivated by the operations of a Turing Machine, a mathematical model of computation that is capable of implementing any computable algorithm. A Turing machine consists of a "tape" with symbols and a "head" that, at each step, moves left or right on the tape, and can read and write symbols in a single tape cell. Therefore, when a Turing Machine processes an input, the tape at each step is a copy of the previous one up to a few changes. Our Turing Programs follow this philosophy by decomposing an algorithmic task into a series of steps. At each step we update a "tape" by copying the previous tape with a few elementary changes. We refer the reader to Figure 1 for the correspondence between Turing Machines and Turing Programs and to Figures 2 and 13, for examples of Turing Programs.

Using the Turing Programs technique, we show that transformers enhanced with the Hard-ALiBi positional encoding (Jelassi et al., 2024)—a recent encoding that achieves state-of-the-art length generalization on copying—are capable of length generalization on a wide range of algorithmic tasks. Our method achieves non-trivial length generalization on addition, multiplication and simulation of SGD steps (see Table 1). Additionally, we show that transformers can be trained to execute random Turing machines, extrapolating from 50 to over 100 input tokens, suggesting that our method can work for general algorithmic tasks. To our knowledge, these are the first results showing non-trivial length generalization on multiplication, and the first attempt to study length generalization on complex algorithms like SGD. We hope that this recipe will be further used to unlock novel length generalization on other algorithmic tasks.

Our key contributions are summarized as follows:

- In section 3, we present length generalization results on multi-digit addition using a Turing Program and Hard-ALiBi positional encoding.

- In section 4, we present the Turing Program framework in full generality and its connections to Turing machines. Additionally, we theoretically prove that transformers can implement Turing Programs, constructing

a RASP program (Weiss et al., 2021) simulating Turing machines.

- In section 5, we demonstrate that Turing Programs are general and lead to novel length generalization results in unexplored algorithmic tasks: multiplication by 1 or 3-digit operand, SGD for linear regression, performing sequences of arithmetic operations, and Turing Machine simulation.

## Related work

Length generalization remains an important challenge for large language models as underlined in several works (Delétang et al., 2022; Dziri et al., 2024; Hupkes et al., 2020; Schwarzschild et al., 2021; Zhang et al., 2022). Despite their advanced reasoning capabilities, Transformer-based large language models struggle to process longer sequences than they were trained on (Anil et al., 2022). The main approaches for improving length generalization focus on changing the positional encoding and optimizing the data format.

**Positional encodings for length generalization.** Shaw et al. (Shaw et al., 2018) were early to notice that the weak length generalization of Transformers was due to the choice of absolute positional encoding. Following this, many alternatives were proposed to replace the absolute positional encoding: relative positional encodings, which focus on the relative distances between tokens (Dai et al., 2019); and weighted attention mechanisms in place of position embeddings (Chi et al., 2022; Jelassi et al., 2023; Li et al., 2023; Press et al., 2021; Raffel et al., 2020). These alternatives showed substantial improvements in length generalization on natural language processing tasks. On the other hand, (Kazemnejad et al., 2024) found that a causal language model with no positional encoding can length generalize better than some of these specialized positional encodings on algorithmic tasks. In this work, we apply the Hard-ALiBi positional encoding (Jelassi et al., 2024), that achieved state-of-the-art length generalization on the specific task of copying, to more general algorithmic tasks.

**Data formatting for length generalization.** A wide range of data formatting methods have been introduced to achieve length extrapolation in algorithmic tasks. Scratchpad and Chain-of-Thought formats were proposed to learn arithmetic either through finetuning or in-context learning (Anil et al., 2022; Zhou et al., 2023). When training from scratch, some other proposed techniques to improve length generalization on addition include: reversed formatting and random space augmentation (Shen et al., 2023), adding padding to the sequence (Jelassi et al., 2023), and setting index hints in front of each digit (Zhou et al., 2023). Closer to our work, several

works (Anil et al., 2022; Dziri et al., 2024; Hu et al., 2024; Kazemnejad et al., 2024; Lanchantin et al., 2024) report that training or finetuning a model on scratchpad data does not yield any significant length generalization improvement. In our work, we demonstrate that length generalization is possible using a combination of a particular scratchpad variant and a favorable positional encoding. Additionally, we develop Turing Programs, a novel scratchpad strategy that is general and may be applied to achieve length generalization on any algorithmic task.

**Neural networks and Turing Machines.** Many prior works designed architectures inspired by Turing Machines (Dehghani et al., 2018; Graves et al., 2014; Kaiser & Sutskever, 2015). From a theoretical perspective, some works proved the Turing completeness of RNNs (Chen et al., 2017; Siegelmann & Sontag, 1992), transformers (Bhattamishra et al., 2020; Chung & Siegelmann, 2021; Pérez et al., 2019; Wei et al., 2022a; Merrill & Sabharwal, 2023) and even linear next-token predictors (Malach, 2023) under a wide range of assumptions. Lastly, another line of work characterizes the computational model that Transformers express: (Weiss et al., 2021) introduce RASP, a human-readable programming language that can be implemented by transformers, (Lindner et al., 2024) show how human-readable programs are compiled into transformer models and other works (Giannou et al., 2023; Jojic et al., 2023) study how transformers can emulate computer programs. Closer to our work, (Zhou et al., 2024) hypothesize that Transformers can length generalization on any algorithmic task that may written as a "simple" RASP program. In this work, we construct a simple RASP program that generates Turing Programs to simulate arbitrary Turing machines.

## 2. Setting

In this section, we present the length generalization problem and some instances where it appears. Then, we discuss scratchpad prompting (Nye et al., 2021), a technique that lets the model generate solution steps before producing the final answer. Finally, we introduce various positional encoding methods and discuss their implications on length generalization.

### 2.1. Length generalization

Many sequence modeling tasks have problem instances of different lengths. Shorter instances are often easier to state, process and handle, and require less compute to find the answer. By contrast, longer instances are more challenging to parse and require more compute to solve. Reasoning tasks such as multi-hop reasoning, program execution, deductive reasoning, and theorem proving fit in this category.

Algorithmic reasoning tasks consist of inputs that are se-

quences of tokens describing the task (e.g. addition, multiplication) and outputs that are the corresponding solutions. We assume that the language model is allowed to generate (many) intermediate tokens before outputting the answer. Then formally, the *length generalization* problem consists of training a language model on inputs of length $\leq L$ and solving problems of length $> L$ at test time.

### 2.2. Scratchpad

It has been shown in prior work that the performance of LLMs on algorithmic tasks can be greatly improved by generating step-by-step solutions instead of immediately outputting the final answer (Wei et al., 2022b). Among the multiple methods described in the literature, we focus on the scratchpad method (Nye et al., 2021). Given an algorithmic task, this method encodes the intermediate steps of the algorithm as text and trains the model to emit them to a buffer that is referred to as the "scratchpad".

Nye et al. (Nye et al., 2021) showed that scratchpad finetuning can be used to achieve strong in-distribution performance on execution based tasks such as code execution and computing polynomials. They also report modest length generalization results on integer arithmetic. The limitation of scratchpad training for length generalization is further highlighted in (Anil et al., 2022; Dziri et al., 2024; Hu et al., 2024; Kazemnejad et al., 2024; Lanchantin et al., 2024).

In this paper, we revisit the use of scratchpad training to achieve length generalization on algorithmic tasks. We begin with the observation that the scratchpad technique can be realized as an iterative sequence of copying operations, where at each iteration the input is slightly modified. Building on previous works showing that with the right positional encoding, transformers can achieve length generalization on the copying operation (Jelassi et al., 2024) , we hypothesize that combining the scratchpad technique with a favorable positional encoding can unlock length generalization capabilities. We verify this hypothesis in section 3 and section 5, but first we review various choices of positional encoding.

### 2.3. Positional encodings

The inability of transformers to extrapolate to longer sequences has been primarily attributed to the positional encoding (Shaw et al., 2018; Shen et al., 2023). In this section, we review the different positional encoding schemes and in section 3, we report their length generalization performance. We review here specific choices for positional encodings that are known to perform well for length generalization, and discuss additional encoding schemes (such as absolute and relative positional encodings) in Appendix A.

**No Positional Encoding (NoPE).** Decoder-only models with causal attention, as shown by (Haviv et al., 2022),

acquire positional understanding, without explicit positional encoding. (Kazemnejad et al., 2024) shows that a model without positional encoding extrapolate better than those with specialized positional encodings on some algorithmic tasks.

**ALiBi.** (Press et al., 2021) introduces this additive positional encoding where the bias function follows $b(i, j) = -r|i - j|$, where $r > 0$ is some hyperparameter. This scheme has led to state-of-the-art length generalization on natural language tasks. However, (Jelassi et al., 2024) notices that it struggles at length generalization on the copy task and hypothesize that it is due to the slow decay of $r$.

**Hard-ALiBi.** (Jelassi et al., 2024) introduce Hard-ALiBi, an additive positional encoding where the bias satisfies $b(i, j) = -\infty$ for $j \leq i - m$ and $b(i, j) = 0$ for $j > i - m$, for some hyperparameter $m > 0$. Intuitively, with this hard thresholding, tokens can only attend to the $m$ closest tokens. Different heads may have different values of $m$ and some heads use $m = \infty$ which corresponds to softmax attention with no positional embedding at all (allowing for propagation of global information). The authors demonstrate empirically that models equipped with the Hard-ALiBi positional encoding achieve remarkable length generalization on the copy task. In this work, we use the Hard-ALiBi position encoding to enable length generalization on algorithmic tasks as we show below.

# 3. Length generalization on addition

```
Input:
4 3 2 4 + 1 3 9

Target:
<scratch>
4 3 2 4 + 1 3 9
4 3 2 e + 1 3 j (1,3)    # added 4 + 9 = 3 carry 1
4 3 c + 1 d (0,63)       # added 2 + 3 + 1 = 6 carry 0
4 d + b (0,463)          # added 3 + 1 = 4 carry 0
e + ^ (0,4463)           # added 4 + 0 = 4 carry 0
4 4 6 3
</scratch>
```

*Figure 2.* Turing Program for addition, text in comments is not part of the input.

In this section, we address the length generalization problem for addition. We first review prior results on this problem and describe the techniques used in these works. We then demonstrate that Transformers trained with Turing Program scratchpads and Hard-ALiBi positional encoding achieve good length generalization performance, extrapolating from length-50 to length-100 addition. This is a remarkable improvement over previous length generalization results using the "vanilla" scratchpad technique (e.g. (Nye et al., 2021)),

which showed weak length generalization performance. As mentioned, there is a long list of works that focus on length generalization on addition (see Appendix B for a complete review). Notably, (Zhou et al., 2024) report somewhat better length generalization results compared to our results. However, we note that these results rely on particular choices for the formatting of the input and the output, which are "tailored" for the task of multi-digit addition.

## 3.1. Length generalization on addition with Turing Programs and Hard-ALiBi

In this section, we present our Turing Program scratchpad strategy for addition and report length generalization results.

### 3.1.1. EXPERIMENTAL SETUP

**Data.** We adopt the scratchpad format and write all the steps into one sequence, where steps are separated by a separator token. Figure 2 shows our scratchpad strategy for getting length generalization on addition If not specified otherwise, our token space is of size 24 and made of $\mathcal{V} = \{0, \ldots, 9, +, a, \ldots, j, \char`\^, <|\mathrm{BOS}|>, <|\mathrm{EOS}|>, <|\mathrm{SEP}|>\}$. All the digits are sampled uniformly as follows: we first sample the length of each operand (between 2 and $L = 50$) and then independently sample each digit. Finally, we "pack the context" with i.i.d. sequences during training, i.e. we fill the context with multiple independent samples of the task (similarly to (Zhou et al., 2023)). We set the training context length to 500. At test time, we evaluate our models using a sliding window: we generate tokens until the training context length (500) is filled, and then each additional token is generated by feeding the context of the most recent 500 tokens, effectively dropping all past tokens[1]. This way, we are able to generate very long sequences of tokens without training or evaluating on long context windows. To evaluate the accuracy at a given length, we test the model's output on 288 examples. We report the accuracy of exactly matching the desired output.

**Model and Training.** Our base model is a 150M parameter Transformer with $L = 12$ layers, a $D = 1024$ hidden size, feedforward layer with a hidden dimension of 4096 and $H = 16$ attention heads. The backbone of our model is based on the GPT-NeoX architecture (Black et al., 2022). We pick a context length of 500 tokens. We use the AdamW optimizer (Loshchilov & Hutter, 2017) to train the model with a weight decay value of 0.1 and no dropout, for 200,000 steps. The learning rate schedule incorporates an initial 100-step linear warm-up, followed by a linear decay, starting at `7e-5`.

---

[1]For efficiency reasons, once we reach the context length we advance the "window" by 20 tokens.

**Hard-ALiBi positional encoding.** Similarly to (Jelassi et al., 2024), we use $M$ masked heads and $(H - M)$ NoPE heads. In the masked heads, we respectively set the hyperparameter $m$ to $1$, $2$,... and $M$. We swept over $\{3, 4, 5, 6, 7, 8\}$ and found that $M = 6$ is the best choice.

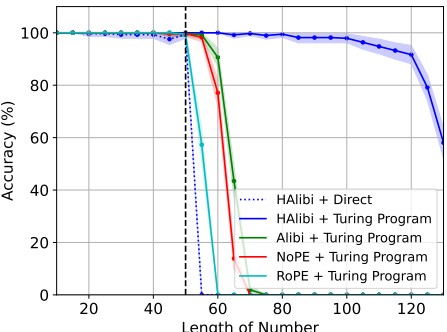

*Figure 3.* Comparison of different positional encodings and data formats for length generalization on addition. We see significant extrapolation to longer sequence lengths with Hard-ALiBi and scratchpad. The shade shows the 95% confidence intervals.

### 3.1.2. RESULTS

In Figure 3 we show the length generalization performance of transformers trained to perform multi-digit addition using the scratchpad described above. We compare the performance of different choices of positional encodings, as well as comparing to the performance on addition without scratchpad (directly outputting the answer).

We observe that by using Hard-ALiBi together with scratchpad, transformers are able to generalize well beyond the length of the training examples. In particular, the Hard-ALiBi model achieves a 98% accuracy at length 100. As shown by Figure 11 in Appendix C, the model also length generalizes well when the operands are of different lengths. In, Figure 8 in Appendix C, we analyze the robustness of length generalization performance to different choices of initialization seed. We observe that, while there is significant variance in performance when testing on longer sequences, our method is more robust compared to prior results (as reported in (Zhou et al., 2024)). Finally, we plot the performance of alternative data formats, such as the "index hints" from (Zhou et al., 2023), with various positional encodings under the same experiment conditions in Figure 12 of Appendix C.

## 4. Turing Programs

In section 3, we showed that Transformers with Hard-ALiBi trained on a specific choice of scratchpad format can length generalize to sequences that are $2\times$ longer. On closer inspection, each line in the scratchpad in Figure 2 is a slightly

modified copy of the previous one where a few elementary changes are applied, e.g. removing one digit for each operand and updating the intermediate result/carry. Since Hard-ALiBi yields robust length generalization on copying, this may explain why we achieve better extrapolation than previous works that trained their models with scratchpad.

In this section, we generalize this approach and claim that every algorithmic task can be written as a sequence of *modified copy* operations: i.e. copy operations with small and localized modifications. Such decomposition follows immediately from the standard construction of a Turing Machine, a universal model of computation. We therefore refer to this scratchpad strategy as a *Turing Program*. We start this section introducing the standard definition of a Turing Machine, and then present Turing Programs, our scratchpad strategy for achieving length generalization on any algorithmic task. Lastly, we present our main theoretical result: Transformers can implement Turing Programs over long sequences of inputs.

### 4.1. Background: Turing Machines

A Turing Machine (Turing, 1950) is a computational model that consists of an infinite tape[2] with *cells*, a head that can read from a cell, write to a cell and move left or right over the tape, and a set of rules which direct the head based on the symbol it reads and the current state of the machine. More formally, a Turing Machine is defined as follows.

**Definition 4.1.** A Turing Machine is specified as a quadruple $T = (Q, \Sigma, s, \delta)$ where: 1) $Q$ is a finite set of states, 2) $\Sigma$ is a finite set of symbols, 3) $s \in Q$ is the initial state and $f \in Q$ is the final state, 4) $\delta$ is a transition function determining the next move: $\delta \colon (Q \times \Sigma) \to (\Sigma \times \{L, R\} \times Q)$.

At the first iteration, the machine is set to state $s \in Q$, the head is on the first (leftmost) cell of the tape, and the input is written on the tape from left to right. At each iteration, the head is on the $i$-th cell in the tape, is in state $q \in Q$ and reads the $i$-th symbol on the tape $\alpha$. Then, if $\delta(q, \alpha) = (\alpha', D, q')$, the head writes the symbol $\alpha'$, moves in the direction $D \in \{L, R\}$, and the machine changes its state to $q'$. If the machine reaches the state $f$, it stops, and its "output" is written on the tape.

Turing Machines are a powerful model for solving algorithmic tasks since (a) the framework is *universal* i.e. it is possible to write any algorithmic task in the Turing Machine formalism, (b) Turing Machines can solve a wide range of algorithmic problems—ranging from simple arithmetic to

---

[2]We assume that the tape is unbounded from the right side, but bounded from the left. Namely, there are infinitely many cells to the right of the input, but no empty cells to the left. This is computationally equivalent to a tape that is infinite from both sides.

determining whether a number is a prime (Agrawal et al., 2004)—in a polynomial number of steps. In the next section, we show how to use the Turing Machine formalism to obtain a novel scratchpad strategy that unlocks length generalization on any algorithmic task.

### 4.2. Turing Programs: a universal scratchpad strategy for length generalization

The left panel of Figure 1 represents the simulation of a Turing Machine and shows how the state, the head and the tape evolves with time. Note that at each time step, the state of the tape is a copy of the previous tape with a few elementary changes such as a move of the head, an edit of a single symbol and a change of state.

The steps in a Turing Machine simulation are similar to a scratchpad strategy where each string is a copy of the previous one with a few modifications. Therefore, we claim that for any algorithmic task that can be solved by a Turing-computable algorithm, there is a corresponding scratchpad strategy for solving this problem (as demonstrated in the right panel of Figure 1). We refer to this novel scratchpad strategy as *Turing Programs*.

Turing Programs decompose an algorithmic task into a series of intermediate reasoning steps. Each step is a "tape" that maintains the state of the machine, and the next step is a copy of the previous tape with a few elementary changes, such as trimming of digits and update of carry/intermediate result as in the case of addition and multiplication (see Figures 2 and 13) or update of the parameters in the case of SGD on linear regression (see subsection 5.2). In section 5, we show how to use Turing Programs to unlock novel length generalization results on challenging algorithmic tasks.

### 4.3. Theory: Turing Programs in RASP

To further motivate the use of Turing Programs to achieve length generalization on arbitrary algorithms, we prove that transformers can implement Turing Programs over long sequences of inputs. In particular, we show that Turing Programs can be implemented in RASP (Weiss et al., 2021), a programming language that was suggested as an abstract description of the operations of a transformer. Following (Zhou et al., 2023), we use a restricted version of RASP that does not allow direct index operations, as (Zhou et al., 2023) hypothesized that RASP programs with index arithmetics may not length generalize[3]. Therefore, our result should be viewed as a length-generalization-friendly construction of a transformer that can execute (most) Turing Programs (and hence, can simulate most Turing machines).

---

[3]Our RASP program does not follow all the restrictions of the RASP-L language suggested in (Zhou et al., 2023), as we do not restrict the tokens to have `int8` values.

To avoid index operations, we leverage the $n$-gram hashing mechanism suggested by (Jelassi et al., 2023) as a basis for the copying ability of transformers. In their construction, copying a string from the input was achieved by storing a sequence of $n$ preceding tokens ($n$-gram) at each position, and iteratively retrieving the next token after the uniquely matched $n$-gram. Our Turing Program construction is very similar, except that instead of copying a string from the input, we copy the next state of the Turing machine as computed from the previous string. As in the construction of (Jelassi et al., 2023), our RASP program is limited to operating on inputs that have no repeated $n$-grams (i.e., no sequence of $n$ tokens appears twice in the input), which can be guaranteed with high probability for uniformly random sequences of tokens of length $\leq \exp(n)$. Additionally, we require that the Turing machine does not generate repeated $n$-grams when processing the input, and that all the operations of the Turing machine are applied in-memory[4]. Under these assumptions, we get the following result:

**Theorem 4.2.** *Let $T$ be a Turing Machine s.t. 1) $T$ does not generate repeated $n$-grams and 2) $T$ operates in-memory. Then, there exists a RASP program $P$ of size (number of lines) $O(n)$ s.t. for every input $x$ without repeated $n$-grams, $P$ correctly simulates $T$ for $\exp(n)$ steps.*

We give the full code for the construction of such RASP programs in Appendix D.

## 5. Length generalization on other algorithmic tasks

Building upon the encouraging length generalization results on addition from section 3 and the Turing Programs framework from section 4, we show that Transformers enhanced with Hard-ALiBi may achieve robust length generalization on complex algorithmic tasks. We show that our framework achieves length generalization on multiplication by 1-digit and 3-digit operands, on SGD applied to linear regression, on performing sequences of arithmetic operations, and finally, on next-state prediction of a random Turing Machine.

### 5.1. Multiplication by a fixed-length operand

**Prior work.** Multiplication is known to be a challenging task for length generalization and very few works report positive length generalization results on this task. On pretrained models, (Zhou et al., 2023) shows that elaborate prompting techniques slightly improve the length generalization

---

[4]Namely, we assume that the head of the Turing machine does not go beyond the input sequence. We believe that this restriction may be removed at the cost of constructing a more complex RASP program. While this may seem like a limiting restriction, we note that this limitation can be easily mitigated by padding the input with random tokens.

of Codex on $(n \leq 3)$-multiplication. (Dziri et al., 2024) show that even fine-tuned GPT-3 struggles with performing 3-digit multiplication. On randomly intialized networks, (Lee et al., 2023) show that models can learn in-distribution the 2-digit multiplication in a sample efficient way using scratchpad. (Shen et al., 2023) shows that with padding and reversed products it is possible to perfectly learn in-distribution 12-digit multiplication. (Jelassi et al., 2023) focuses on 3-digit multiplication and shows that when training on $(5 \times 3)$-digit-multiplication and adding a few examples of $(35 \times 3)$-digit-multiplication, the model length generalizes to $(35 \times 3)$-digit-multiplication. In summary, prior work mainly focused on in-distribution learning of multiplication and did not manage to obtain length generalization results.

**Data setup.** Our experimental setup is similar to the one in section 3. We focus on multiplication by a fixed-length operand, i.e. $(n \times k)$-digit-multiplication where the first operand has variable length $n$ and the second operand always has a fixed length $k \in \{1, 3\}$ across all examples. We adopt the scratchpad format and write all the steps into one sequence, where steps are separated by a separator token. The Turing Program for multiplication is described in Figure 13 of Appendix E. Our token space is similar to the token space used in Section 3, using a $*$ symbol instead of $+$ and using an additional separator token $\sim$. All the digits are sampled uniformly as follows: we first sample the length of the first operand (between 2 and 50) and then independently sample each digit. The remaining details of the training/test protocols are similar to those in section 3.

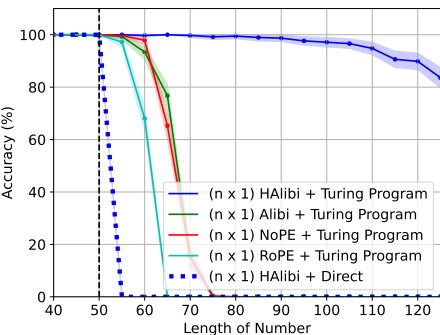

*Figure 4.* Comparison of different positional encodings and data formats for length generalization on $(n \times 1)$-digit-multiplication using the same hyperparameters. The shade shows the 95% confidence intervals.

**Results.** Figure 4 reports our length generalization results on $(n \times 1)$, while the plots for $(n \times 3)$ multiplications are shown in Figure 9 of Appendix C. We obtain robust length generalization by a factor $\times 2$ (from 50 to 100-digit numbers) on $(n \times 1)$ and $(n \times 3)$ multiplication. We note that, up to length 100, $(n \times 1)$ and $(n \times 3)$ multiplication perform

roughly the same ($(n \times 1)$ has accuracy 97.1% and $(n \times 3)$ has accuracy 96.8%), which demonstrates the generality of our Turing Programs framework. Both results are achieved with $M = 7$ masked heads and peak learning rate 0.0003. The head numbers were again chosen by sweeping over candidate numbers as before while the learning rates were chosen from the candidate set $\{\texttt{7e-5}, \texttt{e-4}, \texttt{3e-4}\}$.

### 5.2. SGD on Linear Regression

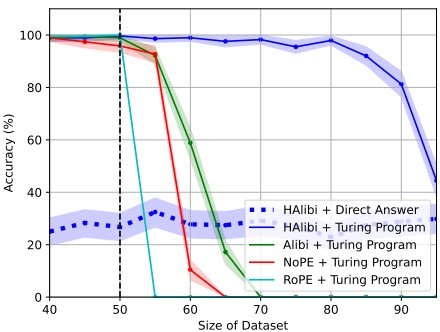

*Figure 5.* Length generalization on running the SGD algorithm, varying the number of examples.

In this section, we train a model to perform SGD and demonstrate its ability to length generalize. While in previous examples we varied the number of digits in the operands, here we instead vary the number of examples.

**Problem Description.** Let $D = \{(\vec{x}_i, y_i)\}_{i=0,...,n-1}$ with $\vec{x}_i \in \mathbb{R}^2$ and $y_i \in \mathbb{R}$ be a dataset of size $n$. Given initial weights $\vec{w}_0 \in \mathbb{R}^n$, we can obtain the final weight $\vec{w}_{n-1}$ by performing gradient descent: $\vec{w}_{i+1} = \vec{w}_i - \lambda \nabla_{w_i}(y_i - \vec{w}_i \cdot \vec{x}_i)^2$, where $\lambda$ is the learning rate. For our experiment, we pick $\lambda = 0.5$ and $\vec{w}_0 = 0$.

**Tokenization and Data.** We divide the interval $[-1, 1]$ into 200 discrete tokens $\{\boldsymbol{a}_0, \boldsymbol{a}_1, ..., \boldsymbol{a}_{199}\}$. As an input, the model receives a sequence of $n$ examples, each encoded as two input coordinate and one output (label) value. The model then needs to compute the iterates of the SGD algorithm when processing the data examples, starting from the last data point, and output the resulting weight vector $\vec{w}_{n-1}$. A detailed description of the Turing Program for solving SGD is detailed in Appendix E.2.

**Results.** Unlike previous experiments, where we report accuracy w.r.t. exact string matching, here we allow the network to err by two quantization unit, counting any output that is within $2/100$ from the ground-truth output (in $\ell_\infty$ norm) as correct. In other words, we disregard errors that may occur to differences in quantization of the real-valued iterates of SGD. As shown by the blue curve in Figure 5,

training the transformer to perform SGD on dataset of sizes $n \le 50$ generalizes with accuracy $> 95\%$ to datasets of size $n = 80$. Our Hard-ALiBi model has $M = 7$ masked heads, a context length of $600$, and was trained with peak learning rate `7e-5` for $400,000$ steps with a batchsize of $16$. For comparison, we also trained a model to directly compute the final answer as shown by the red curve in Figure 5. We observe that training the model to immediately output the answer significantly degrades its performance.

### 5.3. Sequences of Arithmetic Operations

In this seciton, we train a model to perform sequences of Python arithmetic operations as shown below.

```
x1 = 17
x2 = 11
x3 = 17
x4 = 15
y = x4 - x3
y = y * x3
y = y - x1
print(y)
```

We vary the number of arithmetic operations before the final `print(y)` and test whether the models can generalize to programs with more number of operations than those seen in training.

#### 5.3.1. IMPLEMENTATION

In our implementation, we always assign values to 20 initial variables `x1`,..., `x20`. Then we append $n$ calculation steps, where $n$ is a random length sampled from $\{1, 2, ..., 50\}$ during training. The only operations involved are $+$, $-$, and $*$. We also make sure all initial and intermediate values are in the range of 0 to 20. For tokenization, our vocabulary consists of variable names, digits 0 through 9, arithmetic operators ($+$, $-$, and $*$), and various delimiters. We perform one line of calculation for every step of the Turing Program, while copying the initial value assignments and the unfinished calculation steps to the next block. An example of this Turing Program can be found in Appendix E.3.

#### 5.3.2. RESULT

As seen in Figure 6, training the transformer on data with number of calculation steps $n \le 50$ generalizes with accuracy $> 97\%$ to those with number of calculation steps $n = 80$.

### 5.4. Turing simulations

In this section, we test the length generalization of transformers trained to predict the next state of an arbitrary, randomly generated, Turing machine. Our experimental setup is similar to the one in section 3 except for the data as detailed below.

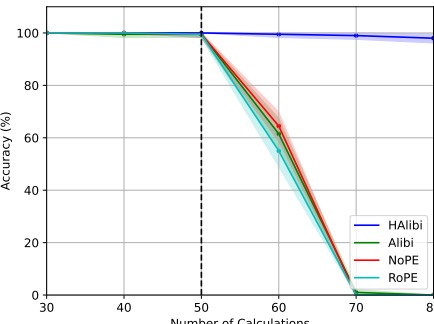

*Figure 6.* Length generalization performance on sequences of arithmetic operations.

**Data setup.** We first sample a random Turing Machine $T$ with 5 states, 15 input symbols and a random transition function (i.e., for every pair of state and symbol we randomly draw a triplet of state, symbol and move-direction). During training, each input example is generated as follows: we randomly choose an input sequence length $L$ between 2 and 50, then randomly choose $L$ tokens, a random position for the head and a random state for the machine. At each step of training, we generate in an online manner a batch of size 16 of Turing simulations from $T$ and focus on learning 1-step prediction: given the input tape, the model has to generate the output of the transition function followed by the next state of the tape. At test time, we evaluate the model on tapes of length $L \ge 50$. Further details are in Appendix F.

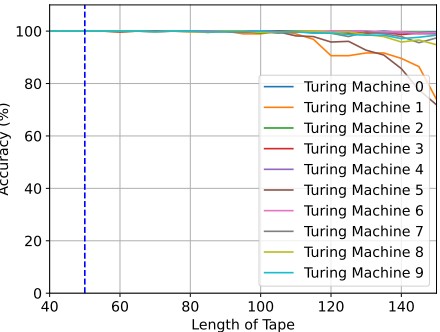

*Figure 7.* Length generalization performance on 10 different randomly generated Turing machines.

**Results.** Figure 7 shows that transformers enhanced with Hard-ALiBi predict almost perfectly the 1-step Turing Machine transition of tapes that are $2\times$ to $3\times$ longer than those seen during training. Trained with a peak learning rate of `7e-5`, the models have $M = 8$ masked heads and a context length of 450. This experiment suggests that transformers may length generalize on *arbitrary* Turing Programs[5]. However, this admittedly does not imply that transformers can

---

[5]We note that, formally, the experiment demonstrates the ability of transformers to learn in the "average case", but does not rule

successfully execute Turing Programs for multiple steps, as accumulating errors might cause the programs to fail. That said, we note that in many cases we get length generalization with virtually zero error, suggesting that multiple steps of the machine can be execute while still maintaining accurate performance. The performance of different positional encodings and data formats for Turing simulation can be found in Appendix C. We observed that both directly outputting the answer and using alternative positional encodings significantly degraded the performance of length generalization.

## 6. Discussion and Limitations

Studying and improving the length generalization abilities of transformers on algorithmic tasks has been the focus of various recent works. In parallel, it has been established experimentally that the ability of language models to solve algorithmic tasks is greatly enhanced when allowing them to use scratchpad/CoT data. Additionally, recent theoretical works demonstrate that transformers can use CoT to simulate arbitrary algorithms (Merrill & Sabharwal, 2023), establishing that they are computationally "universal". These results motivate us to study whether transformers are universal *learners*, able to learn from examples to execute arbitrary algorithms. Since algorithms are typically defined over arbitrary sequence lengths, we use length generalization as a measure of whether the model has learned the *true* algorithm. To establish this, we use the key observation that transformers can length generalize on the copying operation. Since executing an algorithm can be implemented as a sequence of "smart" copy operations, the copying ability of transformers can be leveraged to achieve non-trivial length generalization performance on a wide range of algorithmic tasks.

That said, we acknowledge that our work still falls short of demonstrating that transformers can robustly length generalize on *any* algorithmic task. In some of our results, the extrapolation to longer sequence length is not robust, and degradation in performance may appear shortly after moving out-of-distribution. Additionally, our results rely on potentially very long and cumbersome CoT data, in a way that is not necessarily useful for real-world applications of language models. Thus, we view our results as theoretical evidence that length generalization is possible, and leave the development of more practical and robust methods for real-world length generalization to future work.

out the possibility that some "worst case" Turing Programs have much more restricted length generlization.

## Impact Statement

This paper presents work whose goal is to advance the field of Machine Learning. There are many potential societal consequences of our work, none which we feel must be specifically highlighted here.

## Acknowledgements

This work has been made possible in part by a gift from the Chan Zuckerberg Initiative Foundation to establish the Kempner Institute for the Study of Natural and Artificial Intelligence. EM is supported by The William F. Milton Fund from Harvard University.

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

## A. Additional Positional Encodings Review

**Absolute Positional Encoding (APE).** APE consists in maintaining a positional vector $p_i$ for each position $i$. This vector is either predefined via a sinusoidal function (Vaswani et al., 2017) or learned (Devlin et al., 2018). Then, $p_i$ is added to the token embedding $e_i$ before being processed by the Transformer. Prior work observed that this positional encoding does not generalize well to longer sequences in both natural language (Press et al., 2021) and algorithmic tasks (Jelassi et al., 2023; Kazemnejad et al., 2024).

**Additive Relative Positional Encoding (RPE).** (Shaw et al., 2018) were the first to integrate positional encodings at the level of each attention layer (instead of doing it at the input level). (Raffel et al., 2020) built upon this approach and added scalar biases to pre-softmax logits as follows:

$$A = XW_Q(XW_K)^\top + B, \tag{1}$$

where $X$, $W_Q$, $W_K$ are the input and query and key weight matrices. The bias matrix $B \in \mathbb{R}^{n \times n}$ is induced by some positional encoding function $b \colon \mathbb{N}^{*2} \to \mathbb{R}$. For instance, the T5 relative positional encoding (Raffel et al., 2020) set $b(i, j) = f(i - j)$, where $f$ is some function. Most of the subsequent positional encodings such as ALiBi (Press et al., 2021), Kerple (Chi et al., 2022), Randomized Positional Encoding (Ruoss et al., 2023) and Fire (Li et al., 2023) rely on changing the pre-softmax logits and differ in their definition of $b$.

**Rotary Positional Encoding (RoPE).** RoPE (Su et al., 2024) encodes position information in attention logits by applying a rotation transformation to the query and key vectors based on their relative positions. Despite being widely used, RoPE exhibits limited length generalization (Press et al., 2021; Kazemnejad et al., 2024).

## B. Prior results on multi-digit addition

In this section, we summarize the methods proposed by prior work to get length generalization on addition along with their corresponding performance. In what follows, we indicate in red the positional encoding and in green the data format used in these works. We also take as a running example the addition `576+361=937`.

– (Lee et al., 2023) from 7 to 7-digit ($\mathbf{1.0 \times}$). APE + Reversed format. They train their models by reversing each operand as `675+163=739`. Therefore, the causal model that processes information from left to right can start with the least significant digit and proceed to the most significant digit, which corresponds to the algorithm for addition. They do not achieve any length generalization.

– (Kazemnejad et al., 2024) from 8 to 9-digit ($\mathbf{1.125 \times}$): NoPE + Reversed format. They show that a model without positional encoding trained on reversed additions like `675+163=739` outperforms those with specialized positional encodings like T5's relative positional (Raffel et al., 2020) or RoPE (Su et al., 2024).

– (Shen et al., 2023) from 10 to 11-digit ($\mathbf{1.1 \times}$): NoPE + Reversed format + random space augmentation. They introduced random spacing between digits, aiming to alleviate the model's reliance on absolute positional information. Combining this with the reversed format, the running example becomes `6 75+16 3=739`. They show that NoPE Transformers length generalize from 10 to 11 digit-addition.

– (Zhou et al., 2023) from 30 to 45 digits ($\mathbf{1.5 \times}$): NoPE + Index Hints. They define "index hints", a formatting that consists in adding a letter in front of each digit in the addition to indicate their position. For instance, the running example becomes `a5b7c6+a3b6c1=a9b3c7`. This change is applied during training and inference and enables transformers to execute indexing via induction heads (Olsson et al., 2022).

– (Zhou et al., 2024) from 40 to 100 digits ($\mathbf{2.5 \times}$): Fire (Li et al., 2023) + Randomized positional encoding (Ruoss et al., 2023) + Reversed format + Index Hints . They use a combination of two positional encodings: Fire (Li et al., 2023), a additive relative positional encoding that has obtained strong length generalization on natural language benchmarks and Randomized positional encoding (Ruoss et al., 2023): a technique that samples encodings from a range exceeding test-time lengths. The goal is to ensure that Transformers can adapt to larger positional encodings during training and not encounter any OOD encoding at test-time. With reversed format and index hints, the data format looks like `a6b7c5+a1b6c3=a7b3c9`. By using all these modifications, they reach state-of-the-art performance on length generalization for addition. However, these choices seem to be specific to the addition case and hard to transfer to other algorithmic tasks.

# C. Additional experimental results

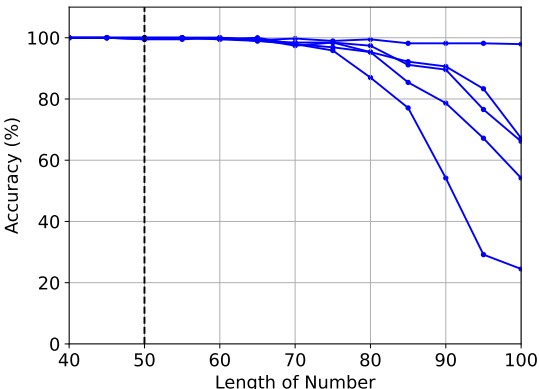

*Figure 8.* Hard-ALiBi with Turing Program, trained to do addition with 5 different initialization seeds. To clarify, the randomness used to plot the 95% confidence intervals in Figure 3 comes from the samples we draw to calculate the accuracy once a seed is fixed, not from different training seeds.

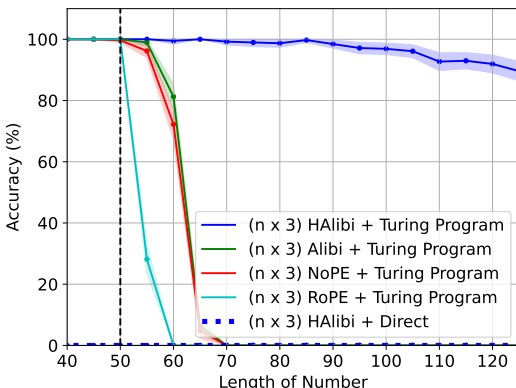

*Figure 9.* Comparison of different positional encodings and data formats for length generalization on $(n \times 3)$-digit-multiplication using the same hyperparameters. The shade shows the 95% confidence intervals.

To the best of our knowledge, (Zhou et al., 2024) achieved length generalization mainly for addition when the two summands had the same length. Our method generalizes even when the two summands have different lengths. For $L_1, L_2 \in \{17, 32, 47, 62, 77, 92\}$, we sampled 96 addition examples where the first summand has length $L_1$ and the second summand has length $L_2$. The accuracy for each combination is shown in Figure 11. We see that it generalizes well beyond the trained distribution ($L_1, L_2 \leq 50$).

## C.1. Comparison with Past Methods

In this section, we show the performance of some of the methods mentioned in Appendix B under our experimental conditions. We consider three data formats:

- Reversed format in (Shen et al., 2023).

- Index hints in (Zhou et al., 2023)

- Index hints + Reversed format in (Zhou et al., 2023)

Moreover, we consider three positional encodings: ALiBi, NoPE, and RoPE. We performed the addition experiments under the exact hyperparameter setting of Figure 3. The results are shown in Figure 12.

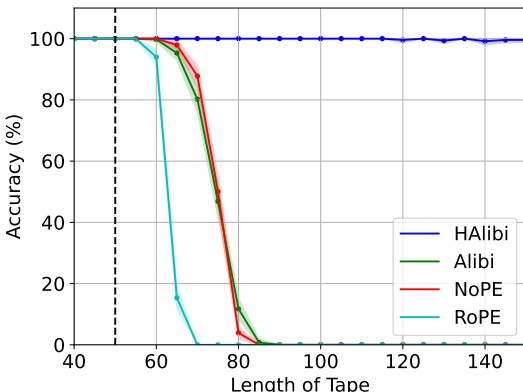

*Figure 10.* Comparison of different positional encodings for length generalization on a randomly generated Turing machine using the same hyperparameters (peak learning rate of $7e - 5$, batch size of $16$, trained for $200,000$ steps).

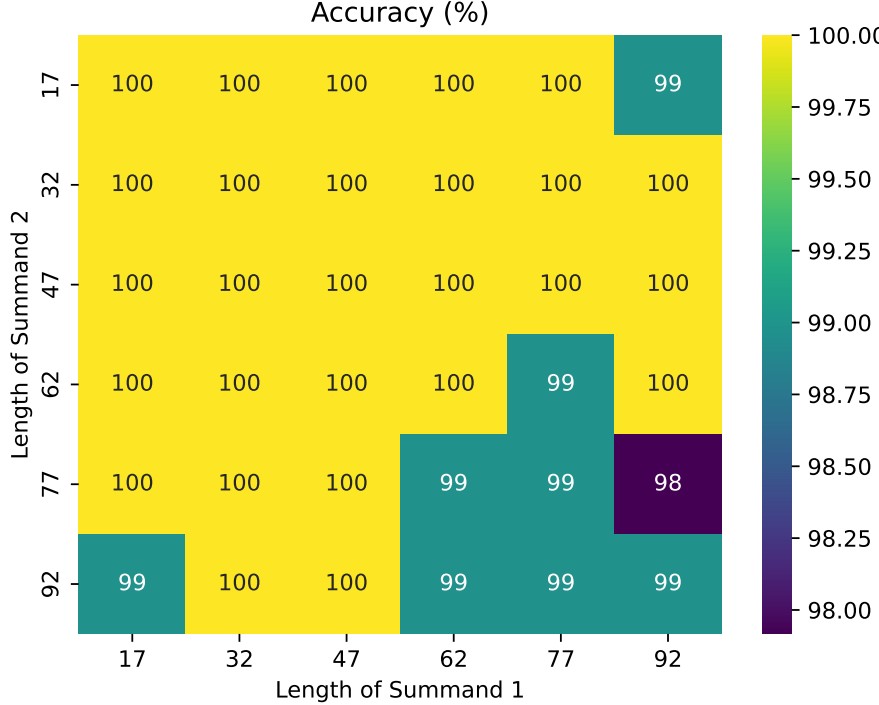

*Figure 11.* Grid displaying the accuracy of our model on addition when changing the length of each operand. We observe that our model is able to generalize on operands with different lengths.

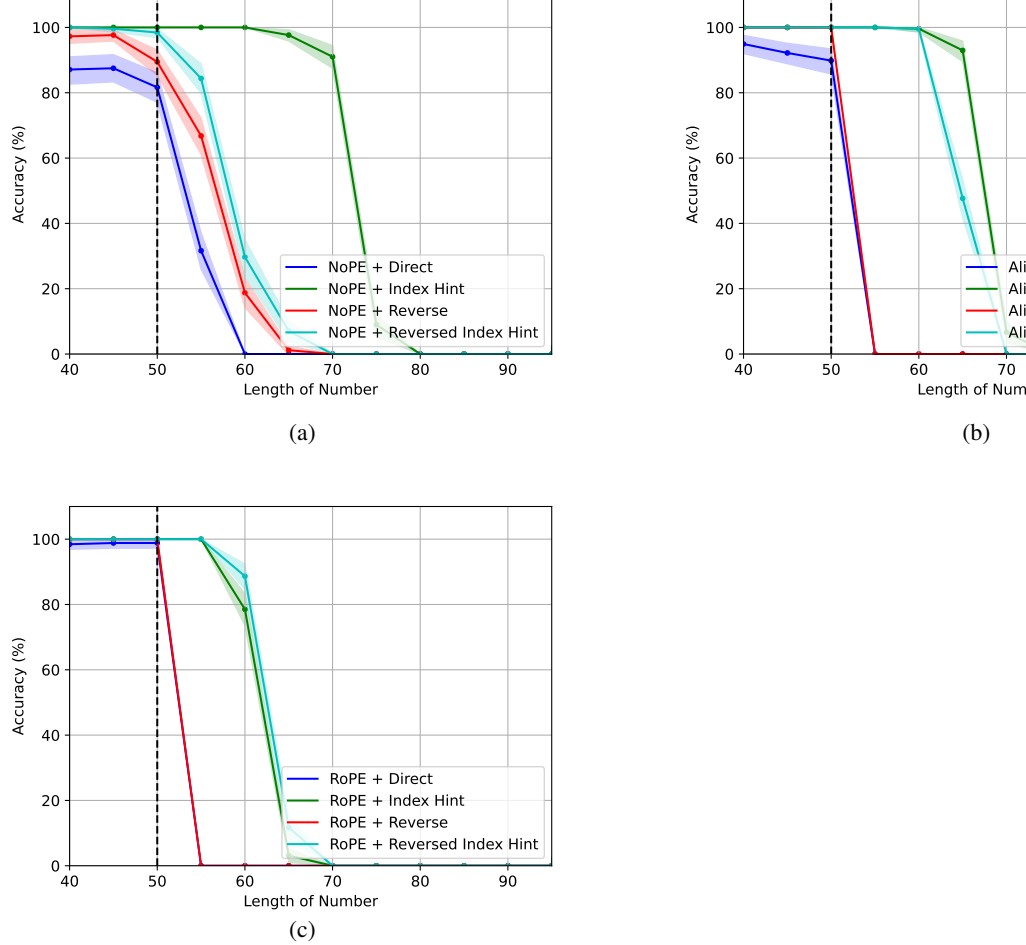

(a)

(b)

(c)

*Figure 12.* Comparison of different positional encodings and data formats for addition. All hyperparameters were held fixed: learning rate of $7e-5$, batch size of 16, and trained for 200k steps.

# D. RASP Turing Programs

## D.1. RASP Python Definitions (from (Zhou et al., 2023))

```python
import numpy as np

def full(x, const):
    return np.full_like(x, const, dtype=int)

def indices(x):
    return np.arange(len(x), dtype=int)

def tok_map(x, func):
    return np.array([func(xi) for xi in x]).astype(int)

def seq_map(x, y, func):
    return np.array([func(xi, yi) for xi, yi in zip(x, y)]).astype(int)

def select(k, q, pred, causal=True):
    s = len(k)
    A = np.zeros((s, s), dtype=bool)
    for qi in range(s):
        for kj in range(qi+1 if causal else s):
            A[qi, kj] = pred(k[kj], q[qi])
    return A

def sel_width(A):
    return np.dot(A, np.ones(len(A))).astype(int)

def aggr_mean(A, v, default=0):
    out = np.dot(A, v)
    norm = sel_width(A)
    out = np.divide(out, norm, out=np.full_like(v, default, dtype=float), where=(norm !=
    0))
    return out.astype(int)

def aggr_max(A, v, default=0):
    out = np.full_like(v, default)
    for i, row in enumerate(A):
        idxs = np.flatnonzero(row)
        if len(idxs) > 0:
            out[i] = np.max(v[idxs])
    return out.astype(int)

def aggr_min(A, v, default=0):
    return -aggr_max(A, -v, -default)

def aggr(A, v, default=0, reduction='mean'):
    if reduction == 'mean':
        return aggr_mean(A, v, default)
    elif reduction == 'max':
        return aggr_max(A, v, default)
    elif reduction == 'min':
        return aggr_min(A, v, default)

def kqv(k, q, v, pred, default=0, reduction='mean'):
    return aggr(select(k, q, pred), v, default=default, reduction=reduction)
```

## D.2. Additional Functions (from (Zhou et al., 2023))

```python
    import operator as op
import numpy as np

# Define comparison operators
equals, leq, lt, geq, gt = op.eq, op.le, op.lt, op.ge, op.gt
```

```
def shift_right(x, n, default=0):
    # shifts sequence x to the right by n positions
    return kqv(indices(x) + n, indices(x), x, equals, default=default)

def cumsum(bool_array):
    # returns number of previous True elements in bool_array
    return sel_width(select(bool_array, bool_array, lambda k, q: k))

def where(condition, x_if, y_else):
    # equivalent to np.where(condition, x_if, y_else)
    x_masked = seq_map(x_if, condition, lambda x, m: x if m else 0)
    y_masked = seq_map(y_else, condition, lambda y, m: y if not m else 0)
    return seq_map(x_masked, y_masked, lambda x, y: x if y == 0 else y)

def mask(x, bool_mask, mask_val=0):
    # equivalent to x*bool_mask + default*(~bool_mask)
    return where(bool_mask, x, full(x, mask_val))

def maximum(x):
    return kqv(x, x, x, lambda k, q: True, reduction='max')

def minimum(x):
    return -maximum(-x)

def argmax(x):
    mm = maximum(x)
    return kqv(mm, x, indices(x), reduction='max')

def argmin(x):
    return argmax(-x)

def num_prev(x, queries):
    # output[i] = number of previous elements of x equal to queries[i], inclusive
    return sel_width(select(x, queries, equals))

def has_seen(x, queries):
    return kqv(x, queries, full(x, 1), equals, default=0)

def firsts(x, queries, default=-1):
    # find the index of the first occurrence of each query[i] in x
    # out[i] := np.flatnonzero(x[:i+1] == queries[i]).min()
    return kqv(x, queries, indices(x), equals, default=default, reduction='min')

def lasts(x, queries, default=-1):
    # find the index of the last occurrence of each query[i] in x
    # out[i] := np.flatnonzero(x[:i+1] == queries[i]).max()
    return kqv(x, queries, indices(x), equals, default=default, reduction='max')

def index_select(x, idx, default=0):
    # indexes into sequence x, via index sequence idx
    # i.e., return x[idx] if idx[i] <= i else default
    return kqv(indices(x), idx, x, equals, default=default)

def first_true(x, default=-1):
    # returns the index of the first true value in x
    seen_true = kqv(x, full(x, 1), full(x, 1), equals, default=0)
    first_occ = seq_map(seen_true, shift_right(seen_true, 1), lambda curr, prev: curr and
    not prev)
    return kqv(first_occ, full(x, 1), indices(x), equals, default=default)

def induct_kqv(k, q, v, offset, default=0, null_val=-999):
    # get value of v at index of: first occurrence of q[i] found in k (if found) + offset.
    # (excludes the last OFFSET tokens of k from matching)
    # null_val is a special token that cannot appear in k or q; used to prevent accidental
```

```
     matches
    indices_to_copy = firsts(shift_right(k, offset, default=null_val), q, default=null_val
    )
    copied_values = index_select(v, indices_to_copy, default=default)
    return copied_values

def induct(k, q, offset, default=0, null_val=-999):
    return induct_kqv(k, q, k, offset=offset, default=default, null_val=null_val)

def induct_prev(k, q, offset, default=0, null_val=-999):
    # A version of induct for negative offsets.
    indices_to_copy = firsts(k, q, default=null_val) + offset
    copied_values = index_select(k, indices_to_copy, default=default)
    return copied_values
```

### D.3. Utility Functions

```
1 def prefix_fill(x, n, value):
2     ones = full(x, 1)
3     no_fill = shift_right(ones, n)
4     return where(no_fill, x, full(x, value))
5
6 def where3(cond, x, y, z):
7     out = where(cond == 0, x, y)
8     return where(cond == 2, z, out)
```

### D.4. Turing Machine Transition Function

```
sep = 0
bos = 1
eos = 2
empt = 3
alphabet = list(range(4, 16))
state_space = list(range(16, 32))

state_transition = {a: {s: np.random.choice(state_space) for s in state_space} for a in
    alphabet + [bos, eos]}
symbol_transition = {a: {s: np.random.choice(alphabet) for s in state_space} for a in
    alphabet}
move_direction = {a: {s: np.random.choice([0, 1]) for s in state_space} for a in alphabet}

def next_state(state, token):
    if token in state_transition.keys() and state in state_space:
        return state_transition[token][state]
    else:
        return 0

def next_symbol(state, token):
    if token in alphabet and state in state_space:
        return symbol_transition[token][state]
    elif token == bos:
        return bos
    elif token == eos:
        return eos
    else:
        return 0

def move(state, token):
    if token in alphabet and state in state_space:
        return move_direction[token][state]
    elif token == bos:
        return 1
    else:
        return 0
```

## D.5. Computation of Next Tape State

```python
def get_next(x, x_left, x_right):
    # compute the next state of head and new symbol, without moving the head
    x_state = seq_map(x, x_left, next_state)
    x_symbol = seq_map(x_right, x, next_symbol)
    x_move_R = seq_map(x, x_left, move)
    is_head = tok_map(x, lambda z: z in state_space)
    is_head_right = tok_map(x_right, lambda z: z in state_space)
    x_next = where(is_head, x_state, x)
    x_next = where(is_head_right, x_symbol, x_next)
    x_move_R = x_move_R & is_head
    return is_head, x_next, x_move_R

def select_move_token(no_head_around, head_left_move_left, head_left_move_right,
   head_right_move_left, head_right_move_right, is_head_move_left, is_head_move_right):
    LEFT_TOKEN = full(no_head_around, 0)
    CUR_TOKEN = full(no_head_around, 1)
    RIGHT_TOKEN = full(no_head_around, 2)
    out = CUR_TOKEN
    out = where(head_left_move_right | is_head_move_left, LEFT_TOKEN, out)
    out = where(head_right_move_left | is_head_move_right, RIGHT_TOKEN, out)

    return out

def move_head(cur_state, right_state):
    is_head, cur_next, move_R = cur_state
    right_is_head, right_next, right_move_R = right_state
    left_is_head, left_next, left_move_R = shift_right(is_head, 1), shift_right(cur_next,
    1), shift_right(move_R, 1)

    no_head_around = (~left_is_head & ~right_is_head & ~is_head)
    head_left_move_left = left_is_head & ~left_move_R
    head_left_move_right = left_is_head & left_move_R
    head_right_move_left = right_is_head & ~right_move_R
    head_right_move_right = right_is_head & right_move_R
    is_head_move_left = is_head & ~move_R
    is_head_move_right = is_head & move_R

    x_sel_move = select_move_token(no_head_around, head_left_move_left,
    head_left_move_right, head_right_move_left, head_right_move_right, is_head_move_left,
    is_head_move_right)
    return where3(x_sel_move, left_next, cur_next, right_next)

def next_tape(x, shift):
    # compute the state of the head, after shifting by some n >= 2
    x_ = shift_right(x, shift)
    x_left = shift_right(x, shift+1)
    x_right = shift_right(x, shift-1)
    x_right_right = shift_right(x, shift-2)

    # compute the next state (before moving the head) for current tape and right tape
    cur_state = get_next(x_, x_left, x_right)
    right_state = get_next(x_right, x_, x_right_right)

    x_next = move_head(cur_state, right_state)

    return x_next
```

## D.6. Hashing Functions

```
MAX_INT = 32
def hash_n_gram(x, n):
    out = x
    before_last_sep = tok_map(x, lambda z: z == 0)
    shifted = shift_right(x, 1)
    for i in range(n):
        shifted_is_sep = tok_map(shifted, lambda z: z == 0)
        before_last_sep = shifted_is_sep | before_last_sep
        to_add = seq_map(shifted, before_last_sep, lambda a, b: a*(1-b))
        # add to hash
        out = seq_map(out, to_add, lambda a, b: b + MAX_INT * a)
        shifted = shift_right(shifted, 1)
    return out

def hash_n_gram_iter(x, n):
    is_sep = tok_map(x, lambda z: z == 0)
    sep_cs = cumsum(is_sep)
    x_hash = hash_n_gram(x, n)
    return seq_map(sep_cs, x_hash, lambda a, b: a + (MAX_INT**n)*b)
```

### D.7. Next-Token Prediction for Turing Programs

```
def next_token_turing(x):
    x_next_tape_2 = next_tape(x, 2)
    x_next_tape_3 = next_tape(x, 3)
    x_next_tape_3 = prefix_fill(x_next_tape_3, 2, empt)
    k = hash_n_gram_iter(x_next_tape_3, 1)
    q = hash_n_gram_iter(x, 1)
    v = x_next_tape_2
    out = kqv(k, q, v, equals, reduction='max')
    return out[-1]
```

## E. Turing Program Descriptions

### E.1. Multiplication

```
Input:
4 3 2 4 * 1 3 5

Target:
<scratch>
4 3 2 4 * 1 3 5
4 3 2 e * 1 3 5 (0540~054,0)    # 4 * 135 = 0540 carry 054
4 3 c * 1 3 5 (0270~032,40)     # 2 * 135 = 0270 carry 032
4 d * 1 3 5 (0405~043,740)      # 3 * 135 = 0405 carry 043
e * 1 3 5 (0540~058,3740)       # 3 * 135 = 0540 carry 058
^ * 1 3 5 (0000~005,83740)      # 0 * 135 = 0000 carry 005
^ * 1 3 5 (0000~000,583740)     # 0 * 135 = 0000 carry 000

5 8 3 7 4 0
</scratch>
```

*Figure 13.* Turing Program for 3-digit multiplication. At each step, we update three information: the head position, the result of the "local" multiplication, the carry and the intermediate result of the "global" multiplication.

### E.2. SGD

We briefly describe here the Turing Program we used in subsection 5.2. Beyond the numerical tokens "a0, a1, a2,... a199", we include tokens "$, d, yp, g , cur , |" to aid the calculation. A typical CoT for a gradient descent then looks like the

following:

$ d a179 a166 , a76 d a80 a145 , a102 d a77 a139 , a103 |

d a179 a166 , a76 d a80 a145 , a102 d a77 a139 , a103 yp a100 g a101 a99 cur a99 a101 |

d a179 a166 , a76 d a80 a145 , a102 yp a101 g a100 a99 cur a99 a102 |

d a179 a166 , a76 yp a100 g a120 a117 cur a79 a85 |

In the above example, the first line provides a dataset of size three where "d a179 a166 , a76" denotes the first example ("a179"and "a166" are the coordinates of $\vec{x}$, "a76" is the value of $y$, and "d" is a token that denotes the start of an example). From the second line onward, we perform gradient descent starting from the last data point, working backward: On the second line, the original dataset is copied, while the "a100" following "yp" is the predicted value of $y$ given the initial weight and the last feature vector "a77 a139", the "g a101 a99" says that $\lambda \nabla_{w_i} ||y_i - \vec{w}_i \cdot \vec{x}_i||$ has value "a101 a99", and "cur a99 a101" means that the current weight after update is "a99 a101". After a example's gradient is calculated, we delete that example.

### E.3. Sequences of Arithmetic Operations

```
x1=16
x2=18
x3=7
x4=9
y=x1-x3
y=x4-y
y=y*x1
-----
x1=16
x2=18
x3=7
x4=9
y=16-7
y=x4-y
y=y*x1
current y=9
-----
x1=16
x2=18
x3=7
x4=9
y=9-9
y=y*x1
current y=0
-----
x1=16
x2=18
x3=7
x4=9
y=0*16
current y=0
```

## F. Turing Programs for Simulating Turing Machines

We use the tokens space $a_1, a_2, \ldots, b_1, b_2, \ldots, s_1, s_2, L, R|, (, ), \sim, \texttt{<|BOS|>}, \texttt{<|EOS|>}, \texttt{<|SEP|>}\}$, where the $a_j$'s are input symbols, the $b_j$'s are symbols substituting the $a_j$'s when the head is pointing to them and $(, ), |, \sim, L, R$ are symbols used to encode the transitions. For instance, the transition $(s_1, a_6, L)$ means that the Turing machines moves to state $s_1$, edits the tape by writing $a_6$ and moves the head to the left.

