# OpenReview forum: "Universal Length Generalization with Turing Programs"
_ICML.cc/2025/Conference — ICML 2025 poster_

### Official Review · Reviewer_6SnS · 2025-02-21

**Overall Recommendation:** 3

**Summary:**

This work proposes Turing Program, which is a CoT strategy that decomposes an algorithmic task into steps mimicking the computation of a Turing Machine. The work showed that by using Turing Programs, they obtain robust length generalization on a range of algorithmic tasks: addition, multiplication and in-context SGD.

**Claims And Evidence:**

Yes, the claims made in the submission supported by clear and convincing evidence.

**Essential References Not Discussed:**

N/A

**Experimental Designs Or Analyses:**

I have checked the experimental designs or analyses.

**Methods And Evaluation Criteria:**

The proposed methods make sense the problem or application at hand

**Other Comments Or Suggestions:**

N/A

**Other Strengths And Weaknesses:**

Weakness:
* **The core weakness is the methodology of the work**. In this work, the authors propose a special data format for algorithmic tasks to improve the length generalization performance. Though the performance is promising, the theoretical analysis may be missed.
* **Need More detailed analysis to support why such data format works for Transformer Length Extrapolation**. After reading the work, I am not sure why such a special data format is related to length extrapolation. Any special explanation?
* **Why the method could be used for language modeling**? The paper claims that the method could be used for any algorithmic task, but we are curious where the Turing Program could be used for language modeling.

Therefore, though the methods work well for length extrapolation, it sill needs more reason to explain why such methods work.

**Questions For Authors:**

N/A

**Relation To Broader Scientific Literature:**

This work propose a special data format for length generalization/extrapolation.

**Theoretical Claims:**

There is no theoretical claims in the work.

---

> ### Author Rebuttal · Authors · 2025-04-01
>
> We thank the reviewer for their feedback. We respond to the main points raised by the reviewer below.
>
> **Theoretical analysis may be missed:** If the reviewer can further explain what is lacking in terms of the theoretical analysis, we would love to further explain and add to the revision. From our perspective, theorem 4.2 is the main theoretical result: we show a Transformer of some fixed size can simulate Turing Programs up to length that is exponential in their size, which suggests that “small” Transformers can execute very long programs. This expressivity result is connected to length generalization because it is a necessary (but not sufficient) condition for length generalization to happen with Turing Programs (not sufficient because gradient descent might not learn this constructed solution). Therefore, we feel it provides theoretical backing for the empirical success we see in algorithmic tasks.
>
> **Why special data format works:** We should have made it clearer why the Turing Program data format is useful for length generalization. When we write out a task in this format, the task gets broken down to two subtasks: 1. modifying the tape content at a single position and 2. copying the tape content. The modification only requires the token at the head position and the positions where the Turing machine state is located, so is relatively independent to the overall length of the input. Copying is length dependent, but a past work (https://arxiv.org/abs/2402.01032) already showed that Hard-ALiBi could achieve length-generalization on copying. Thus, we expect Turing Programs combined with Hard-ALiBi to deliver length generalization results.
>
> We explained this understanding in Section 2.2 of the paper (quoted below for ease of reading), but can add further explanation in the revision.
>
> “We begin with the observation that the scratchpad technique can be realized as an iterative sequence of copying operations, where at each iteration the input is slightly modified. Building on previous works showing that with the right positional encoding, transformers can achieve length generalization on the copying operation, we hypothesize that combining the scratchpad technique with a favorable positional encoding can unlock length generalization capabilities.”
>
> **Could be used for language modeling?:** Our method currently is most suited for algorithmic tasks, i.e. problems that can be decomposed into step-by-step solutions. Our work focuses on algorithmic problems that have closed and known solutions (e.g. arithmetic problems), similarly to many existing works on length generalization (e.g. https://arxiv.org/abs/2402.09371). However, we believe that this method can be adapted to a broader set of natural language problems that are of algorithmic nature, such as mathematical reasoning problems, and leave this study to future work.
>
> We thank you again for the helpful comments on our paper. We are happy to provide further clarification, and would appreciate it if you would raise your score if you believe that your concerns were addressed.

---

> > ### Comment · Reviewer_6SnS · 2025-04-05
> >
> > As the current work is not validated on language modeling and is designed specifically for algorithmic tasks, I am afraid that the evaluation is not adequate. Moreover, the author mentions *Copying is length dependent, but a past work (https://arxiv.org/abs/2402.01032) already showed that Hard-ALiBi could achieve length-generalization on copying*. However, we have to know that AliBI/SWA (Sliding Window Attention) can still achieve cheating length extrapolation, as they actually abandon long-distance tokens for focusing on local tokens, so that they cheat to have length extrapolation ability.
> >
> > **To improve the score, the following is necessary**
> > * **Choice 1: Try on real benchmark but not simulation, whatever the real benchmark is**
> > * **Choice 2: Use LongPPL [1] to evaluate the PPL on the addition, multiplication, and in-context SGD tasks**
> >
> > Reference:
> >
> > [1] Fang, L., Wang, Y., Liu, Z., Zhang, C., Jegelka, S., Gao, J., ... & Wang, Y. (2024). What is Wrong with Perplexity for Long-context Language Modeling?. arXiv preprint arXiv:2410.23771.

---

> > > ### Author Response · Authors · 2025-04-07
> > >
> > > We will work on adding length generalization experiments on real mathematical reasoning benchmarks to the final version of the paper, but since this is a short timeframe we will not have results until the discussion period is over, so we politely ask the reviewer to take this into consideration if possible.
> > >
> > > Our plan is to use datasets like gsm8k. Each block of the Turing Program CoT consists of a copy of the original question and a line of calculation leading to the final answer. We will evaluate this approach on whether the model can generalize to problems that require more steps of calculation than those it has seen in training.

---

### Official Review · Reviewer_Wrzm · 2025-03-13

**Overall Recommendation:** 2

**Summary:**

The paper proposes a new method for designing chain-of-thought supervision for algorithmic tasks, termed "Turing Programs". Essentially, the state of a Turing machine (including the tape, head position, and internal state) before and after each transition are serialized and represented in a chain of thought. The authors explore how training with this trace supervision encoded in chain of thoughts improves length generalization on several tasks: addition, multiplication, and SGD on linear regression. This trace supervision combined with HAlibi positional encodings is shown to exhibit significant (although imperfect) length generalization on these tasks.

The paper also includes a theorem that relates to a constructive demonstration (via RASP) that a Transformer can emulate a Turing machine using the proposed encoding (with several simplifying assumptions).

## Update after rebuttal

I think clarifying the main claims of the paper would be an improvement. I also think if the main claims are related to the empirical performance of the proposed chain-of-thought format, a stronger CoT baselines such as the one proposed would be useful to support the claim.

I like the general idea of the paper, and I think if the pledged changes were implemented well and the empirical results still support the main claims, I would likely update my score to a 3, but it is difficult to verify this given that the pledged changes are somewhat significant, and affect the clarity of and support for the main claims. Therefore I think the paper would benefit from resubmission to a future conference or workshop with the proposed changes. However, I don't want to block acceptance if the other reviewers have a different opinion.

**Claims And Evidence:**

The key claims were a bit unclear to me.

There are specific empirical claims related to training with trace supervision in the form "Turing Programs" improving length generalization over training without such supervision, e.g. for addition and multiplication, which appear to be well supported (although could be strengthened by improving the baselines, e.g. considering other chain-of-thought formats).

However, the title of the paper primes the reader to expect a definition of "Universal Length Generalization", and some related result for "Turing Programs". However, the definition and connection to "Turing Programs" was unclear to me. Maybe these are bit nitpicky but:

1. The paper does not seem to establish new expressivity results for Transformers. "Turing Programs" are a specific convention for chain-of-thought sequences, and therefore do not formally extend the expressivity of Transformers. The expressivity of Transformers with chain-of-thought has been studied by prior work (e.g. the cited https://arxiv.org/abs/2310.07923, but also https://arxiv.org/abs/2406.14197 and https://arxiv.org/abs/2402.12875). A key point of complexity in these results is indexing and attending to evolving register states, as well as issues around finite precision. The proposed RASP program seems to avoid this issue through the non-repeated n-gram restriction, and does not restrict tokens to bounded integers, if I understood correctly. It's therefore not clear how this extends our understanding of Transformer expressivity. If this is a key contribution, it would be good to discuss the result in the context of prior work.

2. The paper does not seem to establish a broad class of new learnability results for Transformers. It is already previously known that training with additional chain-of-thought supervision can improve generalization, especially when such chain-of-thoughts effectively "unroll" some dynamic loop (e.g. https://arxiv.org/abs/2310.16028 and https://arxiv.org/abs/2404.15758). "Turing Programs" are simply one specific convention for representing this information, so it's not clear that the result is categorically novel, even though the authors show that it is empirically effective for several tasks. The "universal" claim seems to relate to the fact that any algorithmic task can in theory be encoded as a Turing machine, however the theoretical results limit the set of Turing machines that can be emulated (e.g. the non-repeated n-gram restriction). Additionally, there is no automatic conversion from a given algorithmic task to a Turing Program. Therefore, it's unclear formally what the claim to "universality" is, and why "Turing Programs" have this property in a way that other schemes for encoding chain-of-thoughts do not. For example, chain-of-thoughts in natural language are also "universal" in the sense that they can be used for any task.

In summary: I think the key claims could be clarified for the reader. If the main claims are simply empirical results for the tasks studied, that is fine but should be clearer. If there is some qualitative property of "Turing Programs" that other chain-of-though conventions lack, this property should be more clearly formalized. If there is some new expressivity result, the difference from prior work should be emphasized.

**Essential References Not Discussed:**

Papers with theoretical results on Transformer decoder expressibility: https://arxiv.org/abs/2406.14197 and https://arxiv.org/abs/2402.12875

This paper discusses emulating Turing machines in a Transformer (with external memory): https://arxiv.org/abs/2301.04589

**Experimental Designs Or Analyses:**

See above concern related to baselines.

**Methods And Evaluation Criteria:**

The tasks seem reasonable, although the baselines could be improved. The authors could compare against other conventions for "unrolling" the underlying computation and representing it in a serialized chain-of-thought.

**Other Comments Or Suggestions:**

nits: Should use \citet in several places.

**Other Strengths And Weaknesses:**

Strengths:

* The paper presents a new scheme for representing unrolled computation traces in a serialized chain-of-thought, inspired by Turing machines.
* The paper shows how training with such traces enables strong length generalization for several tasks, including those where training without trace supervision exhibits minimal length generalization.
* The paper gives a constructive result (via RASP) for how Transformer decoders can emulate a subset of Turing machines.

Weaknesses:

* See confusion around key claims and relation to prior work above.

**Questions For Authors:**

What is the size of `n` for the "non-repeated `n`-gram" constraint of Theorem 4.2? It would be good to formalize this a bit more clearly. Can `n` be chosen, i.e. we just require that there exists some `n` such that there are no repeated `n`-grams? Maybe this is clearer from inspecting the RASP code in the appendix, but would be helpful to clarify in the actual theorem statement.

**Relation To Broader Scientific Literature:**

I think the theoretical claims could be better contextualized in prior work, per above comments.

**Theoretical Claims:**

See above.

---

> ### Author Rebuttal · Authors · 2025-04-01
>
> We thank the reviewer for their feedback. We respond to the main points raised by the reviewer below.
>
> **Key claims of the paper:** we want to emphasize that the key claim of our paper is that Transformers can achieve *length generalization* (generalization to problems longer than the ones observed in the training data) for a large class of problems. That is, we do not argue that we establish novel expressivity results—as you pointed out, the fact that language models can express a large class of functions using chain-of-thought has already been shown in various prior works. Rather, we argue, using a combination of empirical experiments and theoretical observations, that Transformers can extrapolate to longer sequence lengths when provided with chain-of-thought/scratchpad data of a particular format which tracks the step-by-step operation of a “Turing Machine” (the Turing Programs). This connection between the chain-of-thought format and length generalization goes far beyond what has been shown in prior works (including the works that you mentioned), which focus on a narrow set of algorithmic tasks. Instead, our results demonstrate the potential of “universal length generalization” — length generalization for any algorithmic tasks. While we do not establish this formally (due to limitations of our theoretical analysis), we believe that the combination of our extensive experimental results and theoretical insights suggests that Transformers can length-generalize on a far larger class of problems than previously acknowledged, a result that we see as the main novel contribution of our work. Therefore, we believe that our theoretical result should be viewed in the broader scope of the paper, coupled with the experiments on length generalization, and not as an independent expressivity result. Thank you for pointing this out, and we will clarify this in the final version of the manuscript.
>
> **Comparison to other baselines:** throughout the paper, we compare our method to training without chain-of-thought, which displays poor length generalization performance. However, we agree that adding a baseline where we train with another chain-of-thought format can help establish our claims, and we plan to run additional baseline experiments and add them to the final version of the paper. Specifically, we will compare our scratchpad technique to more minimal chain-of-thought, for example one that tracks only the current number and the carry digit in the case of multi-digit addition.
>
> **The choice of n in non-repeated n-grams:** we would like to clarify that Theorem 4.2 holds *for any choice of n.* I.e., for any $n \in \mathbb{N}$ there exists a RASP program of size (number of lines) which grows linearly with the chosen *n*, that satisfies the conditions of the theorem. We will clarify this in the final version of the paper.
>
> **Citations:** we will fix the citations format in the paper.
>
> We thank you again for the helpful comments on our paper. We believe that we answered the main drawbacks raised in the review (in particular, about the key claim of the paper), and would appreciate it if you would raise your score if you believe that your concerns were indeed addressed.

---

### Official Review · Reviewer_L1eA · 2025-03-14

**Overall Recommendation:** 4

**Summary:**

The paper tackles the challenge of length generalization in transformer models—the ability to extrapolate from short training sequences to test sequences longer. The main contribution is Turing Programs, a novel scratchpad strategy inspired by Turing machine computations. In this framework, an algorithmic task is decomposed into a series of intermediate “tape” states, where each step is a slightly modified copy of the previous one. Combined with the Hard-ALiBi positional encoding, this approach enables robust length generalization on several algorithmic tasks, including multi-digit addition, multiplication (with both 1-digit and 3-digit operands), and an in-context simulation of SGD for linear regression. The paper also provides theoretical evidence by showing that transformers can implement Turing Programs via construction in the RASP programming language, thereby establishing a formal connection between the proposed method and Turing machine computations.

## update after rebuttal

The authors' response explains the limitations of their work on real data and clarifies their focus on studying position encoding. I find their explanation sufficiently convincing.

**Claims And Evidence:**

- Claims: The paper claims that using Turing Programs enables transformers to generalize to longer sequences on a variety of algorithmic tasks, achieving near-perfect performance (e.g., 98% accuracy on addition when generalizing from 50—to 100-digit numbers) and that transformers can theoretically implement these programs.
- Evidence: The experimental results on addition, multiplication, and SGD, along with detailed comparisons of different positional encoding strategies, support these claims—the provided theoretical construction (Theorem 4.2) further bolsters the claim of universality.

**Essential References Not Discussed:**

N/A

**Experimental Designs Or Analyses:**

The experimental design is sound and thorough.

**Methods And Evaluation Criteria:**

The chosen methods and evaluation criteria are well-aligned with the problem of length generalization in algorithmic tasks.

**Other Comments Or Suggestions:**

- Gap Between the Assumptions and Real-World Data:
In practice, natural language often contains repeated patterns and more complex structures, which could affect the copying mechanism critical to the proposed Turing Program approach. While the authors acknowledge this gap, a deeper empirical or analytical exploration of how these assumptions might limit performance on actual data would be valuable.

- Focus on Algorithmic Tasks vs. Real Language Tasks:
Real-world language tasks could greatly benefit from these insights. It would be interesting to see future work that adapts the Turing Programs framework to more complex, natural language applications, thereby testing whether the benefits observed in algorithmic tasks translate to these richer, less structured domains.

- Role of Positional Encoding in Length Generalization:
While the experiments clearly demonstrate that the choice of positional encoding (notably Hard-ALiBi) is crucial for enabling robust length generalization, it is important to recognize that other components of the transformer architecture also play significant roles. For instance, the attention mechanism, model depth, and training protocols can influence how well the model generalizes to longer sequences.

**Other Strengths And Weaknesses:**

Strengths:
- The approach is novel, drawing inspiration from Turing machines to design a universal method.
- Empirical results across multiple tasks are strong and convincingly demonstrate improved length generalization.
- The theoretical construction adds depth and rigor to the contributions.

Weaknesses:
- The claim of universality might be overextended given that experiments focus on a limited set of algorithmic tasks.
- Some of the theoretical assumptions (e.g., non-repetition of n-grams) may not hold in more complex or noisy real-world scenarios.

**Questions For Authors:**

See list above.

**Relation To Broader Scientific Literature:**

- This paper extends ideas from previous studies on Hard-ALiBi and related positional encoding strategies and situates its contributions in the context of research on transformer expressiveness and Turing completeness.
- By linking empirical improvements to theoretical constructs (RASP programs), the paper offers a meaningful contribution that advances our understanding of length generalization—a long-standing issue in the literature.

**Theoretical Claims:**

The theoretical claim is solid under the assumptions.

---

> ### Author Rebuttal · Authors · 2025-04-01
>
> We thank the reviewer for their positive feedback. We respond to the main points raised by the reviewer below.
>
> **Universality:** We understand that the word “universal” may not accurately capture the nature of our results, and we are open to removing it from the title if the reviewer thinks this will be adequate.
>
> **N-gram repetition**: We agree that relying on n-gram repetition may be a significant restriction of our theoretical construction, but note that for “random enough” inputs, repeated n-grams become very unlikely for large enough n. Additionally, we suspect that transformers in practice may be in fact utilizing an n-gram matching mechanism (as observed in prior works, e.g. https://arxiv.org/pdf/2402.01032), which means that this limitation reflects a true limitation of transformers and not just a problem in our theoretical construction. In that sense, we believe that our theoretical construction truly captures the nature of the solutions learned by the transformers, including the potential limitation of these learned solutions.
>
> **Gap with real-world tasks and data:** This is a legitimate concern. We want to make two points here:
>
> - For this work, our goal is to do a deep evaluation of how length generalization can arise in language models trained on next-token prediction. Following a large body of prior work (see the many papers on addition, such as https://arxiv.org/abs/2402.09371), we conduct these experiments on synthetic tasks. Therefore, we hope this won’t be considered as a fatal weakness for the paper.
> - There may be applications to math problems that have good algorithmic solutions. Consider the game of 24 (analyzed in https://arxiv.org/abs/2404.03683): you win by manipulating 4 numbers to reach 24. It can be solved by DFS, which can be encoded into the Turing Program format. It would be interesting to see if including Turing Programs of various math problems into the training data mix can improve length generalization performance. We leave this study to future work.
>
> **Role of Positional Encoding in Length Generalization:** This is a good point. We agree that many factors contribute to length generalization performance, but we chose to focus on the choice of positional encoding as it was pointed out to be a key factor in prior works studying length generalization. Controlled experiments on how other variables affect length generalization should be done in future research.
>
> We thank you again for the helpful comments on our paper. We believe that we answered the main drawbacks raised in the review, and would appreciate it if you would raise your score if you believe that your concerns were indeed addressed.

---

### Official Review · Reviewer_RWaS · 2025-03-18

**Overall Recommendation:** 3

**Summary:**

This paper introduces Turing Programs, a novel CoT strategy that improves length generalization on a range of algorithmic tasks. By structuring algorithmic tasks as step-by-step computations resembling a Turing Machine, this method achieves robust generalization across tasks like addition, multiplication, and in-context SGD. The authors also provide theoretical proof that transformers can implement Turing Programs.

**Claims And Evidence:**

Yes. The authors conduct experiments on three algorithmic tasks to demonstrate that transformers can achieve length generalization on random Turing Programs.

**Essential References Not Discussed:**

No

**Experimental Designs Or Analyses:**

Yes.

**Methods And Evaluation Criteria:**

Yes

**Other Comments Or Suggestions:**

The abstract and introduction mention that length generalization is a challenge for current LLMs. Since the transformer used in this paper is relatively small (150M), it would be helpful to briefly discuss the potential application of Turing Programs to larger LMs.

**Other Strengths And Weaknesses:**

This paper is the first results showing non-trivial length generalization on multiplication. The experimental design is generally sound and supports the claim that Turing Programs achieve robust length generalization on three arithmetic tasks. However, since arithmetic problems can be effectively solved by deterministic algorithms or Program of Thoughts methods, this may limit the method's generalization to more complex tasks and weaken its practical applicability.

**Questions For Authors:**

Can the proposed Turing Programs method provide non-trivial performance improvements on real-world QA tasks, particularly in mathematical reasoning (e.g., MATH benchmark)?

**Relation To Broader Scientific Literature:**

The paper builds on prior work in length generalization, scratchpad, and other CoT prompting methods.

**Theoretical Claims:**

I did not find any major issues.

---

> ### Author Rebuttal · Authors · 2025-04-01
>
> We thank the reviewer for their positive feedback. We respond to the main points raised by the reviewer below.
>
> **Only algorithmic problems:** The reviewer is right to point out that currently the result is not immediately practical. For this work, our goal is to do a deep evaluation of how length generalization can arise in language models trained on next-token prediction. Following a large body of prior work (see the many papers on addition, such as https://arxiv.org/abs/2402.09371), we conduct these experiments on synthetic tasks. Therefore, we hope this won’t be considered as a fatal weakness for the paper.
>
> **Larger LMs:** We expect no reason why the same generalization won’t hold when our technique is applied to larger models. We want to make two points:
>
> - As model size grows, we expect the model to perform complex Turing programs better. (e.g. harder arithmetic tasks). It would be an interesting direction of future research to quantify and see how model size and task difficulty (maybe captured by the RASP program size as observed in https://arxiv.org/pdf/2310.16028) affect length generalization results.
> - Turing program may be a way to construct CoT for certain math problems (see the next section), which can be used to train large models.
>
> **Improvement in math reasoning:** Although it is unclear how general QA can be improved by the current iteration of Turing Program, there is certainly application to math problems that have good algorithmic solutions. Consider the game of 24 (analyzed in https://arxiv.org/abs/2404.03683): you win by manipulating 4 numbers to reach 24. It can be solved by DFS, which can be encoded into the Turing Program format. It would be interesting to see if including Turing Programs of various math problems into the training data mix can improve length generalization performance. We leave the study of how to use Turing Programs for more realistic math problems to future work.
>
> We thank you again for the helpful comments on our paper. We believe that we answered the main drawbacks raised in the review, and would appreciate it if you would raise your score if you believe that your concerns were indeed addressed.

---

### Decision · Program_Chairs · 2025-05-01

**Decision:**

Accept (poster)

**Comment:**

This paper introduces Turing Programs, a novel CoT strategy that decomposes an algorithmic task into steps mimicking the computation of a Turing Machine. The main contribution involves showing that Turing Programs enables transformers to generalize to longer sequences on a variety of algorithmic tasks, achieving near-perfect performance (e.g., 98% accuracy on addition when generalizing from 50 to 100 digit numbers) and that transformers can theoretically implement these programs. After the rebuttal, most reviewers are satisfied with the authors’ responses, including the revisions and experiments that have been suggested based on the discussions. We encourage the authors to incorporate these revisions and experiments into the revised version of the paper.